# Patterns of polysubstance use among adults in Malaysia—A latent class analysis

**Wan Shakira Rodzlan Hasani**[1,2]*, **Tania Gayle Robert Lourdes**[1], **Shubash Shander Ganapathy**[1], **Nur Liana Ab Majid**[1], **Hamizatul Akmal Abd Hamid**[1], **Muhammad Fadhli Mohd Yusoff**[1]

**1** Institute for Public Health, National Institutes of Health, Ministry of Health Malaysia, Setia Alam, Selangor, Malaysia, **2** Department of Community Medicine, School of Medical Sciences, Universiti Sains Malaysia, Kubang Kerian, Kelantan, Malaysia

☯ These authors contributed equally to this work.
* shaki_iera@yahoo.com

## Abstract

### Introduction

Polysubstance use is the use of more than one non-prescribed licit or illicit substance at one time. This is a common phenomenon, but little is known about the severity and the various substances used by adults in Malaysia.

### Objective

To determine the pattern of polysubstance use and its associated factors among general adults in Malaysia.

### Methodology

This was a secondary data analysis from the National Health and Morbidity Survey (NHMS) 2019), a cross-sectional population survey with a two-stage stratified random sampling design. A total of 10,472 Malaysians aged 18 years and above participated in this survey. Polysubstance use was defined as concurrent use of more than one substance, either alcohol, tobacco, or drugs (opioids, marijuana, amphetamine/ methamphetamine or kratom). A latent class analysis (LCA) was used to identify the membership of polysubstance groups. The association of class membership with demographic profiles was examined using Multinomial Logistic Regression analysis.

### Results

Fit indices (AIC = 16458.9, BIC = 16443.6) from LCA supported 3 classes solution: Class 1; "moderate-drug" group primarily combination used of tobacco and alcohol (2.4%), Class 2; "high-drug" group using multiple substance including kratom (0.3%) and Class 3; "low-drug" group reporting minimal alcohol and tobacco use or non-user (97.3%). The multinomial model showed young adults (18–40 years) had a higher likelihood of being polysubstance users both for moderate-drug class (OR = 4.1) and high-drug class (OR = 3.9) compared to older age (≥60 years). Chinese (OR = 18.9), Indian (OR = 23.3), Indigenous Sabah &

**Data Availability Statement:** The National Institute of Health, Ministry of Health Malaysia has placed restrictions on sharing the full dataset due to cases involving researchers manipulating the data.

Interested researchers will need to send a formal letter/email to the Director General of Health Malaysia, together with the data request form and proposal, available at (http://iku.moh.gov.my/images/IKU/Document/Form/Borangpermohonandatalatest.pdf). The proposal will be reviewed by the Data Repository team from Biostatistics Sector, National Institute of Health Malaysia to ensure no duplication with other projects that have used the NHMS 2017 data. The data request flow chart is available on the website and can be accessed here (http://iku.moh.gov.my/images/IKU/Document/Form/FlowChartforIPHDataApplication.pdf). The authors also, confirm they did not have any special access privileges that others would not have for the NHMS 2017 data. The contact information of the Director General of Health Malaysia is as follows: mailing address (Director General's Office, Ministry of Health Malaysia, Kompleks E, Aras 12, Blok E7, Presint 1, 62590 Putrajaya, Malaysia); email (anhisham@moh.gov.my); phone number (+60388832545); and fax number (+60388895542).

**Funding:** he author(s) received no specific funding for this work.

**Competing interests:** The authors have declared that no competing interests exist.

Sarawak (OR = 34.6) and others ethnicity (OR = 8.9) showed higher odds of being moderate-drug users than Malays. The greater odds of moderate-drug use for males (OR = 35.5), working groups (OR = 1.5) and low education level group (OR = 3.2).

## Conclusion

Our study highlights patterns and demographics related to the use of polysubstances among adults in Malaysia. These results would help formulate specific prevention programmes for these high-risk groups.

## Introduction

Polysubstance use is defined as the use of more than one non-prescribed licit or illicit substance. Licit substances include alcohol and cigarettes, while illicit substances include marijuana, cocaine, heroin, amphetamine, and methamphetamine [1]. Although most of the studies focused on single substance use, many people especially illicit drug users, use more than one substance and are typically nested in a broader pattern of polysubstance use. Among the many possible combinations of polysubstance use, use of alcohol and other drugs was the most common pattern [2, 3]. However, in Southeast Asian countries, alcohol and tobacco was a common combination of polysubstance use [4]. In Malaysia, the most commonly used substances among adults were tobacco (21.3%), followed by alcohol (11.8%) and illicit drugs (0.5%) [5]. For illicit drug use, the primary drugs used in Malaysia remain opiate-based (heroin and morphine), followed by methamphetamine. In 2016, the numbers of addicts detected among adult (aged 40 and over) are alarming as the numbers increase by 14.7% from previous year [6]. The National Anti-Drugs Agency (NADA) reports that from 2016 to 2020, there was a slight decrease in the number of adult drug addicts in Malaysia [7]. However, this trend does not consider the combination of other substance used including tobacco and alcohol.

The type of polysubstance used may vary in different countries. In Southeast Asian countries, kratom (leaves from the Mitragyna speciosa Korth. trees) is one of the combined substances used by polysubstance users, especially in Malaysia and Thailand. Kratom is an indigenous plant of Southeast Asia and it is reported to have dose-dependent effects like opioids and stimulants [8, 9]. In Malaysia, substance users abuse kratom by mixing the kratom drink with other substances to obtain opioid-like effects. Singh et al. (2014) reported that approximately 15% of the Malaysian population used Kratom to abstain from illicit drugs and alcohol [10].

Empirical research on various patterns of polysubstance use is important to identify the problems in the use of multiple substances. According to the National Epidemiologic Survey on Alcohol and Related Conditions (NESARC), most people with multiple substance use disorders (SUDs) had at least one other co-occurring SUD. The prognosis of multiple SUDs patients was worse than that of single SUD patients [11]. Furthermore, the synergistic interaction of multiple or polysubstance use has been shown to increase the possibility of negative consequences. Previous studies have pointed out that polysubstance use is associated with poisoning or overdose-related death [12–14], an increased risk of poor physical health, risky behaviour, poor response to treatment, and mental health problems [15–17]. Epidemiological research has shown consistent links between polysubstance use and socio-demographic variables. Polysubstance use has been associated with young age [18], lower education [19] and

socio-economic disadvantage [20]. Apart from that, some findings suggest that racial/ethnic differences in the pattern of polysubstance use [21, 22].

Given the huge public health risk of multiple substance use, understanding the pattern of polysubstance use should continue to be a priority not only among adolescents but also for adults. Unfortunately, in Malaysia, the pattern or class of polysubstance use in the adult population is still unclear. Despite having some information pertaining to polysubstance use in Malaysia, existing studies are limited to specific populations such as injection drug users [23, 24], prisoner [25], and men who have sex with men (MSM) [26]. The study on polysubstance use at the national level was only reported among adolescent (aged 13–17 years) [27] and youth (15–40 years) [28]. Therefore, it is necessary to investigate a nationally representative sample of the adult population to provide information for the development of theories explaining the pattern of polysubstance use. These specific types of polysubstance patterns or clusters might respond better to targeted prevention and treatment approaches. In this study, we used Latent Class Analysis (LCA) to identify patterns or latent class membership of polysubstance use among Malaysian adults through a large nationwide representative dataset. Currently, LCA is a method well used in substance use literature to explain the different patterns of substance use [29]. The purpose of this paper is to address the following research questions: (1) What is the class membership of polysubstance use and the prevalence of polysubstance use in each class membership in the general adult's population in Malaysia? and (2) What kind of demographic profile can be used to predict the use of polysubstance in each class membership.

## Material and method

### Data source

This study used data from the National Health and Morbidity Survey (NHMS) in 2019. The NHMS 2019 focused on non-communicable diseases, their risk factors and other health problems, including substance use. This household survey targeted residences in non-institutionalised living quarters (LQ) in Malaysia. Institutional populations such as those staying in hotels, hostels, hospitals, etc., were excluded from this survey. NHMS 2019 is a cross-sectional population survey with a complex survey study design where the sample is representative of the entire Malaysian population. To ensure representativeness, this survey utilised a two-stage stratified random sampling. Population data from the Department of Statistics Malaysia (DOSM) was used as the sampling frame. The stratification involved all states in Malaysia, including the federal territories as the primary stratum. Within the primary stratum, urban and rural areas made up the secondary stratum. Sample selection started at Enumeration Blocks (EB) to the Living Quarters (LQ) and finally to the individual residing in the living quarters. The sample size was calculated using a single proportion formula to estimate prevalence with adjustment for; 1) finite population (based on 2019 projected population Malaysia), 2) design effect (based on previous NHMS 2015 survey), and 3) expected non-response rate of 35%. Thus, the optimum sample size required was 10,544 individuals over the age of 18. Details of the sample size, sampling method, and recruitment procedure have been reported elsewhere [5].

### Measures

**Outcome measurements.** During the NHMS survey, the data for the tobacco module was collected using face-to-face interviews by the trained data collection team. While, the alcohol and drug modules were collected via a self-administered questionnaire. In order to earn their trust and confidence to answer the questions honestly, all necessary steps were taken to ensure that their responses were kept confidential. The respondents were also given the assurance that only the central processing team would have access to all the data, including their responses.

Respondents were neither asked to produce nor verify their identification. They were encouraged to answer the questions by themselves, only referring to the data collectors when terms used were not clear. The respondents were given the option to answer and return the questionnaire immediately, or to return it in a sealed envelope later. Most importantly, they were assured that no legal action would be taken against them if they declared any usage or possession of illicit cigarettes.

*Tobacco.* The questionnaire for tobacco was adapted from the short version of the Global Adult Tobacco Survey (GATS), which had been translated (into the Malay language), pre-tested, and validated among the Malaysian population [5]. Current tobacco use was ascertained by the following question: "Do you currently smoke tobacco on a daily basis, less than daily, or not at all?" Tobacco products included were manufactured cigarettes, hand-rolled cigarettes, kreteks, cigars, shisha, bidis, or tobacco pipes. We defined a current smoker as currently smoking any smoked tobacco product either daily or less than daily. This definition was based on the GATS indicator guideline [30].

*Alcohol.* Alcohol use was ascertained by the following questions: "In the last 12 months, did you consume any alcoholic beverage?" We defined a current drinker as having consumed any alcoholic beverage in the past 12 months based on WHO's definition of alcohol use [31].

*Drug use.* The use of illicit or licit drugs was ascertained by the following questions: "During the past 30 days, did you use these types of drug/substances? 1) opioids (*heroin or morphine*), 2) amphetamines/methamphetamines, 3) marijuana, or 4) kratom". We defined current drug use as taking or using any types of drugs opioids, amphetamines/ methamphetamines, marijuana or kratom in the past 30 days. The definition of drug use during the previous month (last 30 days) were well defined by the World Drug Report [31] and the European drug guidelines [32].

*Polysubstance use.* Polysubstance use was defined as the current use of at least two psychoactive substances in the same time period, either a combination of tobacco and alcohol, tobacco and drugs, alcohol and drugs or combination of all substances.

**Defining predictor variables.** The independent variables used for analysis were sociodemographic variables including gender (male, female), age group (18–40, 41–59, 60 years and above), strata (urban, rural), ethnicity (Malay, Chinese, Indian, Indigenous Sabah & Sarawak, others), education level (no formal education, primary, secondary and tertiary education), occupation (not working, working), marital status (married, unmarried) and monthly household gross income [Bottom 40% (B40), Middle 40% (M40) and Top 20% (T20)]. The household income was calculated based on the self-reported income of each individual and categorised based on the state-specific cut-off for the B40, M40 and T20 categories. The cut-off values for each state were obtained from the Departments of Statistics Malaysia [33].

## Statistical analysis

**Latent class analysis.** The latent class analysis (LCA) approach was used to identify the pattern or group of polysubstance use. LCA is a statistical model used to explore the unobserved heterogeneity in a population and then assign individuals into mutually exclusive and exhaustive types or latent classes based on their pattern of answers to a set of measured variables. It is a type of model-based cluster analysis that generally uses the expectation-maximisation (EM) algorithm for model estimation [34, 35]. In our sample, we used six categorical substance variables as indicators for the latent class model, including current use of opioids (heroin/morphine), amphetamine/methamphetamine, marijuana, kratom, current smoker, and current drinker. This LCA analysis was conducted using R software version 4.0.3 using the "poLCA" package [35]. We estimated a series of class models ranging

from 2 to 5 classes (as we included only six indicator variables, no more than five classes were tested). We then evaluated the models to select the preferred model based on the following fit statistics: Akaike Information Criteria (AIC) and Bayesian Information Criteria (BIC). In general, lower AIC and BIC values indicate a better model fit, as the lower value of the information criterion suggests a better balance between model fit and parsimony [29]. The Likelihood ratio and Chi-Square statistics were also used to assess model fit. Like the AIC and BIC, the aim is to select models that minimise the Likelihood ratio and Chi-Square statistics whilst maintaining a low number of parameters. The larger the value of statistics, the more inefficient the model is to fit the data. After determining the best-fitting model, the posterior probabilities of group membership were used to assign participants to classes. The posterior probabilities refer to the probability of an observation being classified in a given class.

**Multinomial regression.** The multinomial regression analysis was used to assess the association between socio demographic-characteristics with polysubstance groups based on LCA defined classes using STATA software version 15. In this study, a univariable analysis was carried out by testing all the 8 potential predictor variables (age, gender, ethnicity, strata, education, occupation, marital status, and household income) to screen for important independent variables. The variables with $p$-values <0.25 from univariable analysis were included in the preliminary final model (variable selection). The variable selection is the process of "reducing the model" to get the best fit model by including all the candidate variables in the model and repeatedly removing the variables with the highest non-significant p-value until the model contains only significant terms. Hence, the final model was created based on five variables significantly associated at the level of p <0.05 during the final steps of variable selection. Those variables were age, gender, ethnicity, education level, and occupation status. Multicollinearity and interaction were checked accordingly. The overall fitness was checked using a Hosmer-Lemeshow test, classification table, and ROC (receiver operating characteristic) curve for each binary logit model. The findings were presented as crude and adjusted odds ratios with 95% confidence intervals.

**Ethical statement.** Prior to each interview, the purpose of the NHMS survey and methods used during the survey were explained to the respondent, and information was handed out via the participant's information sheet. Those who consented to participate were invited to answer the questionnaire module. This study obtained ethical approval from the Medical Research and Ethics Committee (MREC), Ministry of Health Malaysia (NMRR-18-3085-44207).

## Results

A total of 10,472 Malaysians aged over 18years participated in this survey, giving an individual response rate based on the optimum sample required to be 99.3%. The demographic characteristics of survey respondents are shown in Table 1. The mean age was 32 years (SD = 3.7), with 43% of the sample aged between 18 and 40 years. The sample was predominantly female (54.3%), Malay ethnicity (64.5%), lived in urban areas (60.9%), had a secondary education level (47.7%), was working (56.8%), married (68.3%) and had low household income (68.2%). Table 2 presents the weighted prevalence of substance use. Regardless of polysubstance use, the highest prevalence of single substance use was among tobacco users (22.4%, 95%CI: 20.86, 23.96), followed by alcohol drinkers (11.8%, 95% CI: 10.04, 13.81) and drug users (0.5%, 95% CI: 0.37, 0.79). The total prevalence of polysubstance use was 4.6% (95%CI: 3.69, 5.78) where the combination of alcohol and tobacco (4.21%, 95% CI: 3.29, 5.37) was the highest prevalence among other possible combinations of polysubstance (Table 2).

**Table 1. Socio demographic characteristic of participants (n = 10,472).**

| Socio demographic characteristics | Count[a] | % |
|---|---|---|
| Age (years), mean(±SD) | 31.7 (±3.7) | |
| 18–40 | 4502 | 43.0 |
| 41–59 | 3517 | 33.6 |
| 60 and above | 2453 | 23.4 |
| Gender | | |
| Male | 4785 | 45.7 |
| Female | 5687 | 54.3 |
| Ethnicity | | |
| Malay | 6751 | 64.5 |
| Chinese | 1327 | 12.7 |
| Indian | 662 | 6.3 |
| Indigenous Sabah & Sarawak | 1114 | 10.6 |
| Others | 618 | 5.9 |
| Strata | | |
| Urban | 6380 | 60.9 |
| Rural | 4092 | 39.1 |
| Education level | | |
| No formal education | 644 | 6.2 |
| Primary education | 2379 | 22.8 |
| Secondary education | 4969 | 47.7 |
| Tertiary education | 2425 | 23.3 |
| Occupation | | |
| Not working | 4520 | 43.2 |
| Working | 5944 | 56.8 |
| Marital status | | |
| Married | 7154 | 68.3 |
| Unmarried | 3318 | 31.7 |
| Monthly household gross income | | |
| Bottom 40% (B40) | 6702 | 68.2 |
| Middle 40% (M40) | 2325 | 23.7 |
| Top 20% (T20) | 795 | 8.1 |

[a]Unweighted count

## Latent class analysis

Results of model fitting for each model estimated from LCA are reported in Table 3. Based on these fit statistics, we selected a three-class solution as the best fit for the data. These three-class solutions presented the lowest AIC and BIC with a large improvement of likelihood ratio and chi-square Goodness of fit statistics values over the two-class model. The three-class solution adding another class also did not appear to be meaningful and did not show any significant improvement as the values for AIC and BIC increased after the three-class solution.

The estimated class population share for the three class models is displayed in Fig 1. These are the estimated proportions corresponding to the share of observations belonging to each latent class [35]. The "Predicted class memberships" (posterior probabilities) is another way of estimating the size of the latent classes. Generally, when the values for the "estimated class population shares" and "predicted class memberships" are similar, this is an indication of good

**Table 2. Prevalence of substance use among adults in Malaysia aged 18 years and above (n = 10,472).**

| Types of Substance use | Prevalence (%) (95% CI) |
|---|---|
| Current tobacco use | 22.40 (20.86, 23.96) |
| Current alcohol use | 11.80 (10.04,13.81) |
| Current drug use (total)[a] | 0.50 (0.37, 0.79) |
| Heroin/morphine | 0.03 (0.01, 0.13) |
| Amphetamine or Methamphetamine | 0.10 (0.04, 0.22) |
| Marijuana | 0.10 (0.06, 0.29) |
| Kratom | 0.40 (0.22, 056) |
| Polysubstance (total)[b] | 4.60 (3.69, 5.78) |
| Alcohol + Tobacco | 4.21 (3.29, 5.37) |
| Alcohol + Drug | 0.05 (0.02, 0.12) |
| Tobacco + Drug | 0.43 (0.29, 0.64) |
| Tobacco + Alcohol + Drug | 0.04 (0.01, 0.10) |

Weighted prevalence (adjusted for design weight, non-response rate and population number)

[a] Total current drug use: used any type of drug (including heroin, morphine, amphetamine, methamphetamine, marijuana or kratom) in the past 30 days.

[b] Total polysubstance used: having concurrent use of more than one substance (tobacco, alcohol or drug) at the same time period

model fit. The three-class solution demonstrated almost similar values of estimated and predicted class solutions. Based on posterior probabilities, class 1, class 2 and class 3 contained 2.4%, 0.3% and 97.3% of the sample respectively. Class 1 and class 2 were classified as polysubstance group as they were characterised as having probabilities of using a combination of two or more substances at one period. Fig 2 present the probabilities of the different indicators in each class. The detailed probabilities for each indicator in the LCA model were described below;

- Class 1 was characterised by higher probabilities of tobacco and alcohol use compared to other classes, plus very low probabilities of all types of drug use. We named this class the "moderate-drug" group.

- Class 2 was the smallest sample (0.3%) but had high probabilities of smoking and kratom use, with moderate to low probabilities of all types of illicit drugs, including opiods, marijuana, and amphetamine/methamphetamines, but no probabilities for alcohol use. This multiple substance use class is named the "high-drug" group. Class 2 also demonstrated that our respondents had high probabilities of co-using kratom with other illicit drugs.

**Table 3. Latent class model fit statistics.**

| Classes | AIC[a] | BIC[b] | Likelihood ratio | Chi-square goodness of fit | Maximum log-likelihood | Number of estimated parameters | df |
|---|---|---|---|---|---|---|---|
| 2 | 16458.86 | 16553.20 | 41.60 | 418.75 | -8216.432 | 13 | 50 |
| 3 | 16443.62 | 16550.75 | 12.36 | 24.13 | -8201.812 | 20 | 43 |
| 4 | 16453.77 | 16649.69 | 8.50 | 21.97 | -8199.885 | 27 | 36 |
| 5 | 16467.06 | 16713.78 | 7.79 | 13.39 | -8198.221 | 34 | 29 |

[a]**AIC:** Akaike Information Criteria,
[b]**BIC:** Bayesian Information Criteria

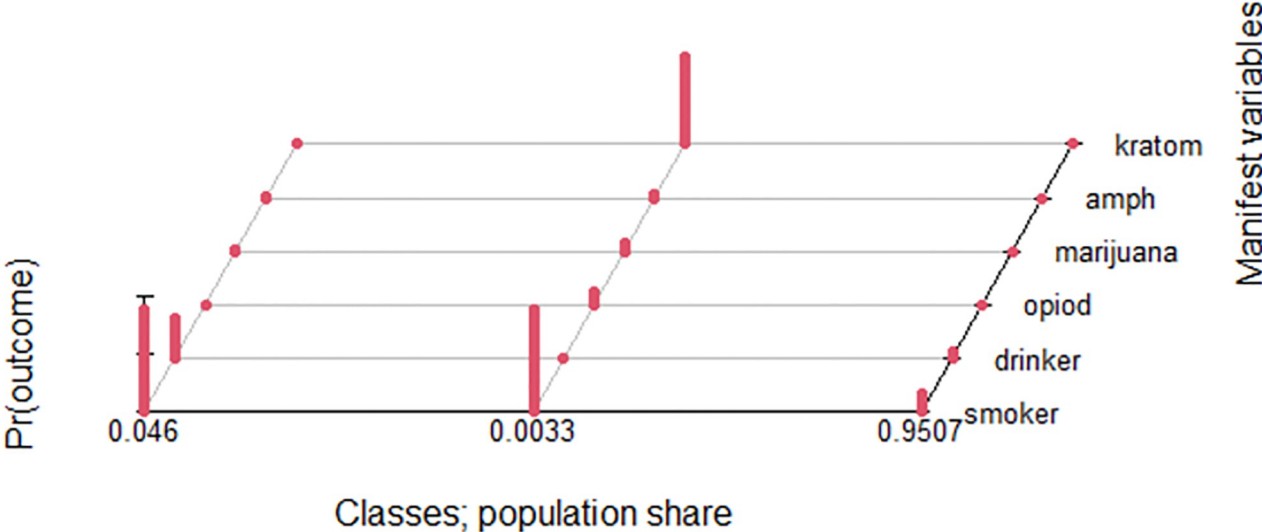

**Fig 1. Estimates class population shares.**

- Class 3 contained the largest class (97.3% of the sample) and was characterised by almost no probabilities of all types of substances and a very low probability of tobacco and alcohol. We named this class the "low-drug" group.

The prevalence of each latent class membership by socio-demographic characteristics is presented in Table 4. Overall, the majority of participants had low-drug use or were non-substance users (class 3). The prevalence of polysubstance use according to latent class 1 (moderate-drug use) and class 2 (high-drug use) was 4.2% (95%CI: 3.25, 5.34) and 0.5% (95% CI: 0.31, 0.66) respectively. Young adults (age 18–40 years) demonstrated a higher prevalence of both moderate-drug use (5.1%) and high-drug use (0.6%) compared with older adults. In terms of gender, males showed a significantly higher prevalence of moderate-drug use as compared to females. No prevalence of high-drug use was estimated for females due to very low cases of multi-drug use among them. The combination of moderate-drugs showed high prevalence among the working group, primary education level, and non-Malay, especially among the Indigenous Sabah and Sarawak. However, Malays demonstrated a high prevalence of high-drug use (0.7%). The prevalence of polysubstance use for both the moderate-drug group and the high-drug group was relatively evenly similar among respondents from urban and rural areas, marital status groups, and household income levels.

## Multinomial regression result

Table 5 presents the odds ratios (OR) comparing all other latent classes (polysubstance class 1 and class 2) to the "low-drug" user (class 3) as reference. In the fully adjusted model, gender, age group, ethnicity, education and working status were significantly associated with polysubstance class 1 (moderate-drug). However, only age group was associated with polysubstance class 2 (high-drug used). In summary, the younger age group (18–40 years) had higher odds of being polysubstance class moderate-drug group [Adjusted OR (AOR) 4.13; 95% CI: 2.45, 6.95) and high-drug group (AOR 3.85; 95% CI: 1.22, 12.12) compared with the older age group (60 years and above). Males had a higher likelihood of being moderate-drug class users than females (AOR 35.51; 95% CI: 17.22, 73.25). Chinese ethnicity (AOR 18.89; 95% CI: 11.16,

## Three Classes Polysubstance use Model

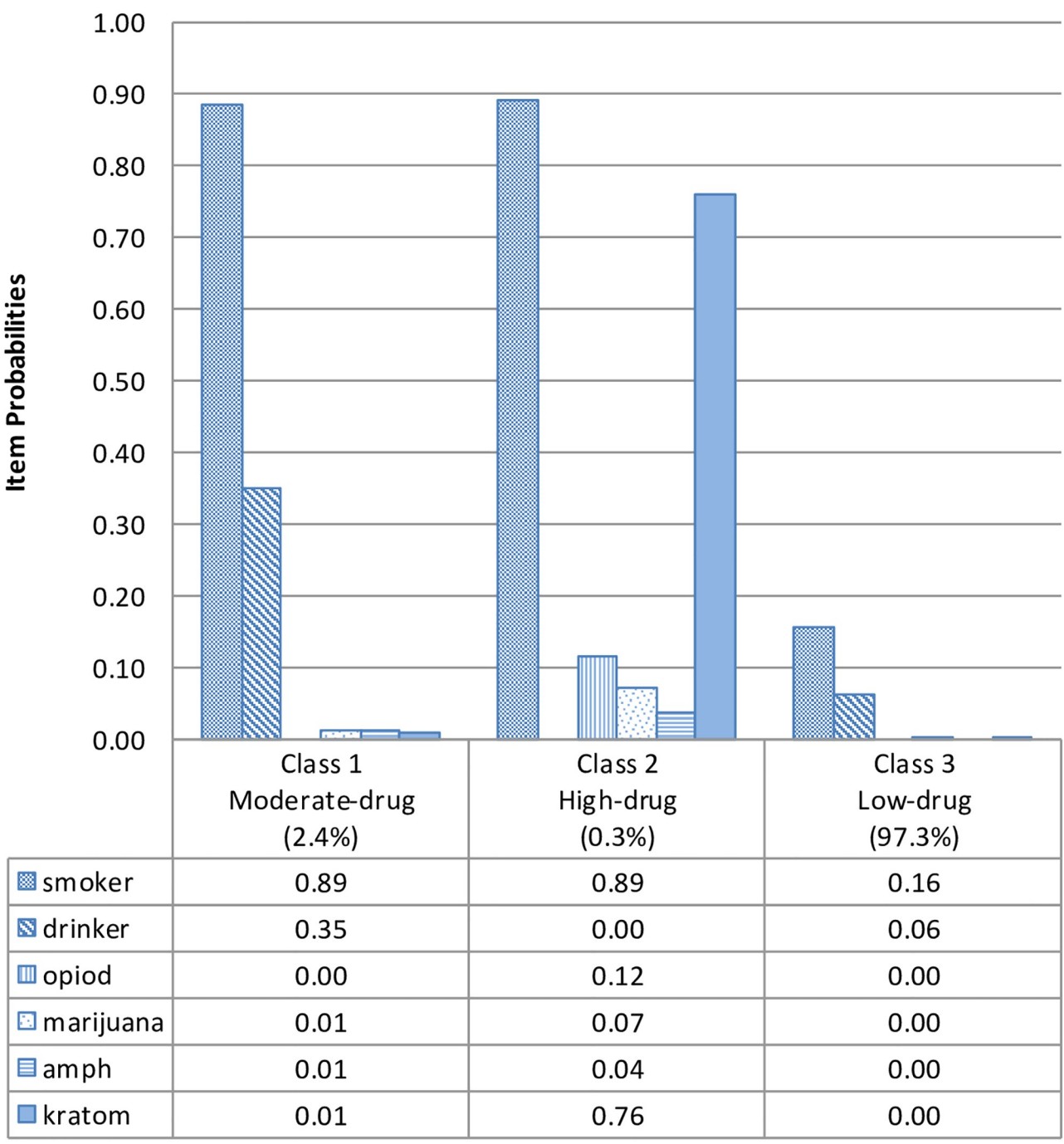

| | Class 1 Moderate-drug (2.4%) | Class 2 High-drug (0.3%) | Class 3 Low-drug (97.3%) |
|---|---|---|---|
| ▦ smoker | 0.89 | 0.89 | 0.16 |
| ◩ drinker | 0.35 | 0.00 | 0.06 |
| ⊞ opiod | 0.00 | 0.12 | 0.00 |
| ⊡ marijuana | 0.01 | 0.07 | 0.00 |
| ▤ amph | 0.01 | 0.04 | 0.00 |
| ▨ kratom | 0.01 | 0.76 | 0.00 |

**Fig 2. Probabilities of substance use indicators in each class in the three-class solution.**

31.97), Indian ethnicity (AOR 23.31; 95%CI: 13.30, 40.86), Indigenous Sabah & Sarawak (AOR 34.55; 95%CI: 21.02, 56.80) and other ethnicities (AOR 8.88; 95%CI: 4.74, 16.62) had a higher likelihood of being a member class of moderate-drug as compared with Malay ethnicity. The odds of being moderate-drug users were also higher among the working group (AOR: 1.50;

Table 4. Distribution of socio-demographic characteristics by latent class membership.

| Socio-demographic characteristics | Prevalence[a] (%) (95% CI) | | |
| --- | --- | --- | --- |
| | Class 1 | Class 2 | Class 3 |
| | Moderate-drug | High-drug | -low-drug |
| Total | 4.2 (3.25, 5.34) | 0.5 (0.31, 0.66) | 95.4 (94.22, 96.31) |
| Age (years) | | | |
| 18–40 | 5.1 (3.73, 7.07) | 0.6 (0.39, 0.96) | 94.2 (92.35, 95.69) |
| 41–59 | 3.6 (2.48, 5.12) | 0.3 (0.16, 0.57) | 96.1 (94.58, 97.25) |
| 60 and above | 1.5 (0.88, 2.55) | 0.1 (0.05, 0.36) | 98.4 (97.32, 99.01) |
| Gender | | | |
| Male | 7.9 (6.12, 10.07) | 0.9 (0.61, 1.28) | 91.2 (89.09, 93.02) |
| Female | 0.3 (0.12, 0.57) | - | 99.7 (99.43, 99.88) |
| Ethnicity | | | |
| Malay | 0.5 (0.28, 0.93) | 0.7 (0.49, 1.07) | 98.8 (98.28, 99.12) |
| Chinese | 6.0 (3.80, 9.43) | 0.1 (0.01, 0.73) | 93.4 (90.46, 96.12) |
| Indian | 7.1 (4.45, 11.20) | - | 92.9 (88.77, 95.52) |
| Indigenous Sabah& Sarawak | 11.8 (9.03, 15.37) | 0.5 (0.14, 1.94) | 87.6 (84.22, 90.40) |
| Others | 8.4 (3.85, 17.38) | - | 91.6 (82.62, 96.14) |
| Strata | | | |
| Urban | 3.8 (2.76, 5.27) | 0.4 (0.26. 0.68) | 95.8 (94.33, 96.83) |
| Rural | 5.4 (3.93, 7.42) | 0.6 (0.33, 0.98) | 94.0 (92.03, 95.54) |
| Education level | | | |
| No formal education | 2.4 (1.29, 4.28) | 0.1 (0.02, 0.84) | 97.5 (95.59, 98.61) |
| Primary education | 6.9 (4.01, 11.55) | 0.3 (0.11, 0.59) | 92.9 (88.25, 95.75) |
| Secondary education | 4.7 (3.45, 6.39) | 0.6 (0.36, 0.97) | 94.7 (93.04, 95.98) |
| Tertiary education | 1.6 (0.95, 2.69) | 0.4 (0.21, 0.83) | 98.0 (96.89, 98.69) |
| Occupation | | | |
| Not working | 1.5 (1.00, 2.30) | 0.2 (0.09, 0.46) | 98.3 (97.50, 98.82) |
| Working | 5.7 (4.37, 7.43) | 0.6 (0.39, 0.92) | 93.7 (91.99,95.05) |
| Marital status | | | |
| Married | 3.9 (2.85, 5.37) | 0.4 (0.24, 0.66) | 95.7 (94.24, 96.76) |
| Unmarried | 4.6 (3.08, 6.89) | 0.5 (0.32, 0.94) | 94.8 (92.60, 96.41) |
| Monthly household gross income | | | |
| Bottom 40% (B40) | 4.9 (3.71, 6.36) | 0.5 (0.30, 0.72) | 94.7 (93.19, 95.84) |
| Middle 40% (M40) | 3.9 (2.12, 7.05) | 0.5 (0.24, 1.24) | 95.6 (92.48, 97.42) |
| Top 20% (T20) | 2.8 (1.57, 4.92) | 0.1 (0.01, 0.58) | 97.1 (94.99, 98.37) |

[a]Weighted prevalence (adjusted for: design weight, non-response rate and estimated number of populations in Malaysian)

95% CI: 1.01, 2.25) versus the non-working group and low education level (no formal education, AOR = 3.23, primary education AOR = 2.55, secondary education AOR = 2.43) versus higher education level.

## Discussion

This study highlights updated or current epidemiological data regarding polysubstance use in the adult population of Malaysia. To our knowledge, this is the first paper to identify the pattern of substance use using LCA among a nationally representative adult population in Malaysia. We identified three distinct classes of substance use among the adult population: those who primarily use a combination of tobacco and alcohol (moderate-drug), those who use

**Table 5. Factors associated with polysubstance use among Malaysian adults from multivariate multinomial logistic regression analysis (n = 10,464).**

| Predictive factors | Univariate Multinomial Logistic Regression | | Multivariate Multinomial Logistic Regression | |
|---|---|---|---|---|
| | Crude OR (95%CI) | | Adjusted OR[a] (95%CI) | |
| | Moderate-drug use | High-drug use | Moderate-drug use | High-drug use |
| | (Class 1) | (Class 2) | (Class 1) | (Class 2) |
| **Age** | | | | |
| 18–40 years | 3.31 (2.14, 5.12)** | 3.25 (1.25, 8.40)* | 4.13 (2.45, 6.95)** | 3.85 (1.22, 12.12)* |
| 41–59 years | 2.15 (1.35, 3.42)* | 1.41 (0.48, 4.14) | 2.40 (1.41, 4.08)* | 1.60 (0.47, 5.44) |
| ≥60 years | 1 | 1 | 1 | 1 |
| **Gender** | | | | |
| Female | 1 | - | 1 | - |
| Male | 36.36 (17.95, 73.66)** | | 35.51 (17.22, 73.25)** | |
| **Locality** | | | | |
| Urban | 1 | 1 | - | - |
| Rural | 1.15 (0.88, 1.49) | 1.57 (0.87, 2.83) | | |
| **Ethnic** | | | | |
| Malay | 1 | 1 | 1 | 1 |
| Chinese | 14.05 (8.48, 23.29)** | 0.14 (0.02, 1.04) | 18.89 (11.16, 31.97)** | 0.18 (0.02, 1.30) |
| Indian | 19.44 (11.34, 33.33)** | 0.29 (0.04, 2.12) | 23.31 (13.30, 40.86)** | 0.32 (0.05, 2.32) |
| Indigenous Sabah&Sarawak | 30.52 (18.95, 49.14)** | 0.71 (0.35, 2.01) | 34.55 (21.02, 56.80)** | 0.76 (0.26, 2.16) |
| Others | 14.59 (8.20, 25.96)** | 0.31 (0.05, 2.24) | 8.88 (4.74, 16.62)** | 0.18 (0.02, 1.42) |
| **Marital Status** | | | | |
| Married | 1 | 1 | - | - |
| Single | 1.31 (1.01, 1.71)* | 1.37 (0.74, 2.51) | | |
| **Education** | | | | |
| No formal education | 1.83 (0.96, 3.49) | 0.42 (0.05, 3.33) | 3.23 (1.56, 6.68)* | 2.10 (0.23, 18.73) |
| Primary education | 2.10 (1.34, 3.29)* | 1.03 (0.41, 2.61) | 2.55 (1.56, 4.17)** | 2.25 (0.82, 6.19) |
| Secondary education | 2.29 (1.53, 3.44)** | 1.38 (0.64, 2.96) | 2.43 (1.58, 3.72)** | 1.48 (0.68, 3.19) |
| Tertiary education | 1 | 1 | 1 | 1 |
| **Occupation** | | | | |
| Not Working | 1 | 1 | 1 | 1 |
| Working | 4.14 (2.93, 5.88)** | 3.52 (1.64, 7.60)** | 1.50 (1.01, 2.25)* | 1.16 (0.50, 2.72) |
| **Income status** | | | | |
| B40% | 1.55 (0.89, 2.68) | 3.85 (0.52, 28.18) | - | - |
| M40% | 1.03 (0.56, 1.89) | 3.09 (0.39, 24.41) | | |
| T20% | 1 | 1 | | |

[a] Multivariate multinomial Logistic Regression was applied for adjusted odd ratio. Interactions were checked and no significant interaction term found. Overall fit the model for each binary logit was checked accordingly: Hosmer-Lemeshow test (logit function 1, p = 0.842, logit function 2, p = 0.281), correctly classified table (logit function 1 = 99.03%, logit function 2 = 97.72%), Area under ROC curve (logit function 1 = 70.0%, logit function 2 = 92.5%).

*p <0.05,

**p<0.001.

multi-drugs including soft and hard drugs (high-drug), and those who are non-drug users or have minimal tobacco and alcohol use (low-drug). This three-class solution from LCA was an interpretability and relevant fit statistics model. Interestingly, the latent classes we identified are somewhat consistent with the findings of previous work that investigated the profile of substance use among men who have sex with men (MSM) in Malaysia, which identified a three-class solution as the best fitting model [26]. Similar to our findings, they also found that the

majority of participants were negligible drug users or non-drug users and detected a small group of samples using hard drugs or multi-drugs.

Our work extends the literature in that we added kratom as one of the indicators for LCA, in addition to other common substance used. We found that there was a high probability of co-use of kratom with other multiple substances (refer to class 2 latent class as visually represented in Figs 1 and 2). This finding was supported by previous studies in Malaysia indicating that people who use drugs (PWUDs) commonly used kratom to abstain from illicit drugs and to manage withdrawal symptoms from both opioid and stimulant use [8, 10, 36]. A study on kratom consumption in the northern areas of peninsular Malaysia also reported that they used kratom to reduce their intake of more expensive opiates like heroin [8]. Although kratom has perceived therapeutic effects, several studies suggested abuse and addiction potential, synergistic toxic effects and fatal interactions with other psychoactive drugs [37, 38]. In Malaysia, kratom is under the jurisdiction of the Poisons Act 1952 [39], where the law stated that any activity involving the possession, sale, use, transportation, processing, importation, and exportation of kratom is prohibited and may result in legal action. However, it remains widely used because the tree grows naturally, is easily accessible and there are no laws that prohibit the cultivation of kratom in Malaysia. Despite not being commonly recognised as a drug in Malaysia, kratom has recently come under scrutiny as an illicit substance. Kratom has reportedly been identified by NADA as one of the drug categories that are abused in this country [40].

In our multinomial regression results, we chose Class 3 (low-drug) as the reference category to distinguish the pattern of polysubstance use from predominantly negligible or non-users. As reported in numerous publications, gender (male) is an important demographic characteristic associated with polysubstance use [19, 22, 41]. This study detected a significant association for the male gender with the moderate-drug group but not for the high-drug group. This may be due to a very low prevalence of multi-drug use among female respondents. In terms of age differences, young adults (18–40 years) demonstrated higher prevalence and risk of being a member of polysubstance use than older adults (≥60 years), for both moderate-drug and high-drug classes. Our findings are in line with the study from the United States that reported the co-use of tobacco and alcohol was highest among young adults and declined with increasing age [42]. As for other polysubstance the risk of use, abuse/dependence, and use of other forms of illicit drugs after the use of marijuana/cannabis declined with increasing age [18]. Another LCA study among the Great Britain population also reported that being young was associated with an increased likelihood of membership of the polydrug use class's wide range and moderate range [41]. There are several explanations for this trend for polysubstance use to decline with increasing age. This includes socio maturity, where increasing age will increase the social maturity of the individual's ability to resist another illicit drug use [43].

Our findings of polysubstance use varying among racial/ethnic groups are consistent with the results from other studies [19, 22, 42, 44]. For example, a study in the United States found that Whites were more likely to drink while Natives American/Alaskan Natives were most likely to smoke and co-use alcohol and tobacco [42]. Our results showed that the co-use of alcohol and tobacco (class 1) was more likely to occur among Indigenous Sabah and Sarawak, followed by Indian and Chinese ethnicity, as compared to Malay ethnicity. In contrast, the Malays ethnic group showed the highest prevalence of high-drug usage (class 2) and the lowest likelihood of consuming alcohol when compared to other ethnic groups. Our study was in line with the findings noted in another epidemiological study in Malaysia reporting similar results, although that study involved adolescents [27]. These findings suggest that the culture and practises of respective ethnic groups might play an important role in predicting substance use. In Malaysia, Malay adults had low prevalence of alcohol use but high prevalence of tobacco use, while Indigenous Sabah & Sarawak showed high prevalence for both alcohol and tobacco

consumption [5, 45]. In this country, Muslims (the majority of whom are of the Malay ethnicity) are forbidden from consuming alcohol, and it is also illegal to sell any alcoholic beverages to them. In contrast, drinking alcohol is a cultural and social obligation among other ethnic groups. For example, indigenous Sabah and Sarawak people consume alcohol during their festivals and social gatherings [46].

The present findings also suggested that adults with low education levels (no formal education, primary education, and secondary education) were more likely to use polysubstance in the moderate-drug group compared to adults with higher education levels (tertiary education). However, no significant association was found between education levels and polysubstance use in the high-drug group. Our findings are consistent with the latent class analysis of polysubstance from European studies, which reported that higher education was associated with a reduced risk of moderate but not high polysubstance classes [19, 41]. They found that there was a reduced risk of moderate polysubstance group membership for participants with an education level beyond secondary education (General Certificate of Secondary Education, GCSE).

Our study also revealed that working adults is a significant predictor for polysubstance use in the moderate-drug (class 1). This is consistent with the epidemiological data in Malaysia, which reported that the prevalence of current smokers and drinkers was higher among working groups, especially self-employed and private employees, compared to those not working [5, 45]. Working conditions may influence addictive behaviours such as exposure to psychological job strain (due to high work demands and low coping skills) that may lead to an increased risk of substance use especially alcohol and tobacco [47–49]. The misuse of alcohol by workers represents an important social policy issue because it has the potential to influence the employee's job performance. For instance, even moderate daily alcohol consumption can lead to absenteeism and smoke breaks among employees that can be considered disruptive as it takes time away from work.

Taken together, the results from the current study suggest that prevention and treatment strategies for polysubstance use, especially for the co-use of alcohol and tobacco, should place a special emphasis on young adults and other high-risk groups such as people from the Indigenous Sabah and Sarawak ethnicity and those who are working. While, the prevention strategies for polysubstance use of multi-drugs, including illicit drugs and kratom, should focus on young adults and people of the Malay ethnicity. Stronger measures to control this drug are warranted and the use of kratom in Malaysia should be further investigated, including the effects of co-use with other drugs.

This study needs to be interpreted with caution in light of several limitations. The primary limitation of this study is the relatively low prevalence rate of multi-drug use such as opioids, marijuana, and amphetamine type stimulants (ATS). Due to the small number of cases of illicit drug use, our study may have been underpowered to identify additional classes and differentiate the pattern of polysubstance classes between hard drugs and soft drugs in the multiple-drug group (class 2) and other potential variables associated with multi-drug use. Our study also used self-reported substance use, where the prevalence rates could be misrepresented. However, past researchers suggested suitable reliability and validity of studies with self-reported substance use [50, 51]. The limitation of this study was the sample recruited from a cross-sectional design of a population-based survey, and it only investigated the socio-demographic predictor factors related to polysubstance use. Although other studies [52–54] have found the association between polysubstance use and psychological distress, as our study extracted the data from a large population-based survey, which is almost similar to other existing population-based studies, mainly cross-sectional, we have no capacity to consider prospectively early life predictors and to describe the association with psychological disorder. Future studies should replicate and extend these findings with a larger sample using a cohort design

and measure the association of polysubstance use with other early life predictors and mental health disorders.

## Conclusion

In summary, the current study provides an illustration pattern of polysubstance use in Malaysia with a three-class solution (moderate-drug, high-drug, and low-drug user). The LCA model also demonstrated a high probability of co-use of kratom together with other illicit drugs in the high-drug classes. Our study emphasises the importance of considering patterns of polysubstance use when addressing demographic risk factors. As noted in our regression model, there was an increased risk of being a high-drug user among the younger age group. While the model for moderate-drug class demonstrated a strong link with males, young adults, non-Malay ethnicity, working groups, and low education level.

## Acknowledgments

We would like to thank the Director-General of Health Malaysia for his permission to publish this article.

## Author Contributions

**Conceptualization:** Wan Shakira Rodzlan Hasani, Hamizatul Akmal Abd Hamid.

**Data curation:** Wan Shakira Rodzlan Hasani, Hamizatul Akmal Abd Hamid.

**Formal analysis:** Wan Shakira Rodzlan Hasani.

**Investigation:** Wan Shakira Rodzlan Hasani.

**Methodology:** Wan Shakira Rodzlan Hasani, Shubash Shander Ganapathy, Muhammad Fadhli Mohd Yusoff.

**Resources:** Wan Shakira Rodzlan Hasani.

**Software:** Wan Shakira Rodzlan Hasani.

**Validation:** Shubash Shander Ganapathy, Muhammad Fadhli Mohd Yusoff.

**Writing – original draft:** Wan Shakira Rodzlan Hasani, Tania Gayle Robert Lourdes.

**Writing – review & editing:** Wan Shakira Rodzlan Hasani, Tania Gayle Robert Lourdes, Shubash Shander Ganapathy, Nur Liana Ab Majid, Hamizatul Akmal Abd Hamid.

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
