## [Decision Letter · Decision Letter 0]

22 Jun 2022

PONE-D-22-03892Patterns of Polysubstance use among Adults in Malaysia – A Latent Class AnalysisPLOS ONE

Dear Dr. Wan Shakira,

Thank you for submitting your manuscript to PLOS ONE. After careful consideration, we feel that it has merit but does not fully meet PLOS ONE’s publication criteria as it currently stands. Therefore, we invite you to submit a revised version of the manuscript that addresses the points raised during the review process.

We look forward to receiving your revised manuscript.

Kind regards,

Mohammad Farris Iman Leong Bin Abdullah, Dr Psych

Academic Editor

PLOS ONE

Journal Requirements:

a) Did participants provide their written or verbal informed consent to participate in this study?

4. Please include a copy of Table 5 which you refer to in your text on page 13.

Additional Editor Comments:

1. Please ensure that the author list and affiliations are correct on the title page of your manuscript, and that your author contributions, competing interests, and financial disclosure are correct as listed below. All of these sections will be indexed in PubMed and published by PLOS ONE as you have written them. Please email plosone@plos.org if any changes to this content need to be made.

Please see here for the full list and definition of contributor roles: http://journals.plos.org/plosone/s/authorship#loc-author-contributions

Reviewers' comments:

Reviewer's Responses to Questions

**Comments to the Author**

1. Is the manuscript technically sound, and do the data support the conclusions?

Reviewer #1: Yes

Reviewer #2: Yes

2. Has the statistical analysis been performed appropriately and rigorously? 

Reviewer #1: Yes

Reviewer #2: Yes

3. Have the authors made all data underlying the findings in their manuscript fully available?

Reviewer #1: Yes

Reviewer #2: Yes

4. Is the manuscript presented in an intelligible fashion and written in standard English?

Reviewer #1: Yes

Reviewer #2: Yes

5. Review Comments to the Author

Reviewer #1: Though the study is timely and highlights the prevalence of poly-substance use among the adult Malaysian population, as well as the use of kratom, the authors must address a few issues based on the review comments before the paper can be accepted for publication.

Introduction

Pg. 2 – paragraph two – most people with multiple substance use disorders (SUDs)….

National Anti-Drugs Agency (NADA) has compiled and described the growing prevalence of poly-drug use among people who use drugs (PWUDs) in Malaysia. Hence, the authors can incorporate some information on poly-substance use in the adult population of PWUDs in Malaysian in the introduction section of their manuscript.

Revision required – In Southeast Asia countries, ketum or kratom (leaves from the Mitragyna speciosa Korth. trees)…

Revision required – Kratom is an indigenous plant of Southeast Asia, and it is reported to have dose-dependent effects as opioid and stimulants.

Kratom (ketum) do not cause/produce hallucinations – please remove the word hallucination from the text. (Obtain opioid-like effects).

Can the authors also provide a rationale as to why the study aimed to determine the patterns of poly-substance use in the adult Malaysian population. What health risks the authors intend to examine?

Methods

Provide an explanation for targeted residence in non-institutionalized living quarters.

How the data collectors ensured that the respondents were willing to disclose their illicit drug use status through the self-administered questionnaire. This is because illicit drug use is illegal in Malaysia, and thus openly self-reporting their illicit drug use status can cause them to experience legal problems with the enforcement authorities in Malaysia.

Did the study also captured other illicit drugs such as ketamine, or other new psychoactive substances (NPSs) used by the respondents.

Were the respondents regular, irregular or those with substance use disorders (SUDs)? Did the authors categorised the respondents as recreational drug users or those with SUDs.

Results

Any reasons why the sample size was larger for urban respondents than respondents from rural setting.

What are the definitions of B40, M40 and T20? Please provide details of the terms used in the methods section of the manuscript.

Discussion

This study highlights an updated or current epidemiological data regarding poly-substance use in the adult population of Malaysia.

Revision needed - This finding was supported by previous studies in Malaysia indicating that people who use drugs (PWUDs) commonly used kratom to abstain from illicit drugs and to manage withdrawal symptoms from both opioid and stimulant use [25,26,34].

What are the authors trying to mean by pure kratom alone? Brewed juice, kratom juice without adulterants, etc.

How can the relevant authorities address kratom cultivation and distribution issues in Malaysia? Please explain what the legal consequences are of using or possessing kratom in Malaysia.

Those who take up marijuana later would be less predisposed to other illicit drug use than those who take it up early – any reference for this.

Additional comments

There are several grammatical errors and poor structuring of sentences. Please look into it.

Reviewer #2: The rationale of the study was not clearly stated. The importance of studying polysubstance abuse and its implication was not clearly stated.

in the methodology, the data collection was not clear whether the data was extracted from a previous record or from interviewing the study population.

The type of substance in table 2 was not specified.

Class 3, was proposed as one of the three Classes of Polysubstance use based on the LCA model analysis. However, it is not clear what is the significance of this class as the use of substances in this class was either non or negligible.

The point about the kratom in the discussion made was beyond the scope and not supported by data from this study.

6. PLOS authors have the option to publish the peer review history of their article (what does this mean?). If published, this will include your full peer review and any attached files.

Reviewer #1: No

Reviewer #2: No

---

## [Author Response · Author response to Decision Letter 0]

28 Jul 2022

Reviewer #1: 

Though the study is timely and highlights the prevalence of poly-substance use among the adult Malaysian population, as well as the use of kratom, the authors must address a few issues based on the review comments before the paper can be accepted for publication.

Introduction

Pg. 2 – paragraph two – most people with multiple substance use disorders (SUDs)….

Feedback: Thank you for your comment. The sentences were edited as suggested.

National Anti-Drugs Agency (NADA) has compiled and described the growing prevalence of poly-drug use among people who use drugs (PWUDs) in Malaysia. Hence, the authors can incorporate some information on poly-substance use in the adult population of PWUDs in Malaysian in the introduction section of their manuscript.

Feedback: Thank you for your comment and suggestion. The introduction section was rephrased accordingly and the statistic information from NADA was added in the introduction. 

Revision required – In Southeast Asia countries, ketum or kratom (leaves from the Mitragyna speciosa Korth. trees)…

Feedback: Thank you for your comment and suggestion. The sentences were rephrased as “kratom (the leaves from the tree Mitragyna speciosa Korth. trees)”

Revision required – Kratom is an indigenous plant of Southeast Asia, and it is reported to have dose-dependent effects as opioid and stimulants.

Feedback: Thank you for your comment and suggestion. The sentences were rephrased as suggested.

Kratom (ketum) do not cause/produce hallucinations – please remove the word hallucination from the text. (Obtain opioid-like effects).

Feedback: Thank you for your comment and suggestion. The word hallucination was removed from the text.

Can the authors also provide a rationale as to why the study aimed to determine the patterns of poly-substance use in the adult Malaysian population. 

Feedback: Thank you for your comment and suggestion.The rational to conduct this study was added in the second last paragraph.

What health risks the authors intend to examine?

Feedback: As this study used the data from cross sectional national wide survey, we could not measure the health risk effect of polysubstance use. This study only intends to explore the pattern of polysubstance use and demographic profile associated with the use of polysubstance. 

However, we described the health effect of polysubstance use including poisoning or overdose-related death, risk for poor physical health, risky behavior, poor response to treatment and mental health problems in the introduction section order to justify the important or public health problem related to polysubstance use.

Methods

Provide an explanation for targeted residence in non-institutionalized living quarters.

Feedback: Thank you for your comment. This sentence was added in the text; “This household survey targeted residence in non-institutionalised living quarters (LQ) in Malaysia. Institutional population such as those staying in hotel, hostels, hospitals, etc. were excluded from this survey.”

How the data collectors ensured that the respondents were willing to disclose their illicit drug use status through the self-administered questionnaire. This is because illicit drug use is illegal in Malaysia, and thus openly self-reporting their illicit drug use status can cause them to experience legal problems with the enforcement authorities in Malaysia.

We do agree that there is an element of bias, whereby the respondent may not be truthful with their answer. However, this is an inherent limitation of all self reported studies, especially when asked questions that are sensitive in nature, eg income, sexuality. 

Feedback: Thank you for your comment. For your information, this NHMS survey had also collected information on other areas, such as alcohol intake, which is forbidden and illegal to be taken by muslims in Malaysia. Furthermore, several studies have also been conducted using self-administered questionnaires, to study issues such as transgender and MSM in Malaysia, all which are also illegal in Malaysia and may subject the respondent to enforcement authorities. All these variables and study findings have been consistent and found to be reliable.

For this study, we have taken all possible measures to ensure confidentiality of the respondents, and to gain their trust and confidence to answer the questions. Respondents are not asked to produce and verify the identification by visual inspection of their identification card, but are allowed to answer the questions themselves. The respondents are given the option to answer the Self-Administered Questionnaire immediately, or to return the questionnaire in a sealed envelope at a later time and date. Furthermore, the respondents are assured that all data, including their responses, can only be seen by the central processing team. [This statement was added in text]. 

Furthermore, the findings of this survey was discussed with the National Anti-Drug Agency, who are agreeable with the findings, even though the obtained prevalence and estimated number of drug users is 4 times higher than the number of drug dependents from the official reports of the National Anti-Drug Agency (source 1). The figures are also fairly similar to the UNODC report for Malaysia (Citation 2). This would be a form of sensitivity analysis of the validity of this study findings. 

Thus, we do agree that there may be some element of underreporting, however this would be the current best estimate of substance abuse in Malaysia, and the figures obtained appear to be consistent with other study findings.

Did the study also captured other illicit drugs such as ketamine, or other new psychoactive substances (NPSs) used by the respondents.

Feedback: This study extracted data from NHMS 2019 surveys, where NHMS did not capture the Ketamine or others new NPSs. However, NHMS collected the self-reported for inhalant use. But we did not include the current inhalant use in LCA model because the prevalence was very low (0.001%) and does not fit with our model.

Were the respondents regular, irregular or those with substance use disorders (SUDs)? Did the authors categorised the respondents as recreational drug users or those with SUDs.

Feedback: Thank you for your comment. Unfortunately, NHMS did not capture for SUDs questions and frequency of drug use due to limited space for questionnaires and time as NHMS is national wide household survey to capture the NCD risk factors and other health problem in a short time. So, we could not categorise the respondents as recreational user and SUD. 

Results

Any reasons why the sample size was larger for urban respondents than respondents from rural setting.

Feedback: The NHMS 2019 covered both urban and rural areas in all 13 states and 3 federal territories in Malaysia where federal territories only consisted of urban enumeration block (EBs). To ensure national representativeness, two stage stratified random sampling was used in NHMS sampling design. The two strata are primary stratum, which made up of states of Malaysia, including Federal Territories, and secondary stratum, which made up of urban and rural strata formed within the primary stratum. So, during sampling design, a total of 5,676 Living Quarters (LQs) were selected from the selected 475 EBs in Malaysia, where 362 (4,320 LQ) from urban and 113 EBs (1,356) from rural were selected. The allocation of samples to the states, urban and rural was done proportionally to the population size. Bigger number of samples were allocated to states with bigger population size. So, this will be the reason why the sample size of urban larger than rural. 

What are the definitions of B40, M40 and T20? Please provide details of the terms used in the methods section of the manuscript.

Feedback: Thank you for your comment and suggestion.This household income statement was added in the method together with definition of other predictors factors variables; “The independent variables used for analysis were socio-demographic variables including gender (male, female), …monthly household gross income [Bottom 40% (B40), Middle 40% (M40) and Top 20% (T20)]. The household income was calculated based on the self-reported income of each individual, and categorized based on state-specific cut-off for B40, M40 and T20 category. The cut-off values for each state were obtained from the Departments of Statistics Malaysia.

Discussion

This study highlights an updated or current epidemiological data regarding poly-substance use in the adult population of Malaysia.

Feedback: Thank you for your suggestion. The sentences were rephrased as suggested.

Revision needed - This finding was supported by previous studies in Malaysia indicating that people who use drugs (PWUDs) commonly used kratom to abstain from illicit drugs and to manage withdrawal symptoms from both opioid and stimulant use [25,26,34].

Feedback: Thank you for your suggestion. The sentences were rephrased as suggested.

What are the authors trying to mean by pure kratom alone? Brewed juice, kratom juice without adulterants, etc.

Feedback: Thank you for your comment. This sentence was removed from text; “Furthermore, there has been no report on the fatalities in Southeast Asia caused by the ingestion of pure kratom alone.”

How can the relevant authorities address kratom cultivation and distribution issues in Malaysia? Please explain what the legal consequences are of using or possessing kratom in Malaysia.

Feedback: Thank you for your comment. The legal statement of using kratom in Malaysia was added in the text. 

Those who take up marijuana later would be less predisposed to other illicit drug use than those who take it up early – any reference for this.

Feedback: Thank you for your comment. This sentence was removed from the text.

Additional comments

There are several grammatical errors and poor structuring of sentences. Please look into it.

Feedback: Thank you for your comments. The English language editing was done accordingly.

Reviewer #2: 

Introduction.

The rationale of the study was not clearly stated. The importance of studying polysubstance abuse and its implication was not clearly stated.

Feedback: Thank you for your comment and suggestion. The rational to conduct this study was added in the second last paragraph.

Method.

in the methodology, the data collection was not clear whether the data was extracted from a previous record or from interviewing the study population.

Feedback: Thank you for your comment. This study used secondary data from NHMS2019 survey. The statement for extracted data was added. To make it clear, we also added the subtopic “Data source” under methodology. In this method, we described the method for data collection used during NHMS survey; “During NHMS survey, the data for the alcohol and drug module were collected via a self-administered questionnaire while, for tobacco module, the data was collected using face-to-face interviews conducted by the trained data collection teams.

Results.

The type of substance in table 2 was not specified.

Feedback: Thank you for your comment. The type of substance was added at the footnote of table 2.

Class 3, was proposed as one of the three Classes of Polysubstance use based on the LCA model analysis. However, it is not clear what is the significance of this class as the use of substances in this class was either non or negligible.

Feedback: Thank you for your comment. We do agree with your statement. Thus, we decided to change the name for all class including class 3;

Class 1: from “Tob+Alcohol” to “moderate-drug” group. Because this group primarily combination used of tobacco and alcohol but very low probabilities of all types of drug use.

Class 2: from “multi-drug” to “high-drug” group. This group had moderate to low probabilities of all types of illicit drugs, including opiods, marijuana, amphetamine/ methamphetamines and high probabilities of smoking and kratom use with no probabilities for alcohol use.

Class 3: from “non/negligible” to “low-drug” group. This group almost no probabilities of all types of substances and a very low probability of tobacco and alcohol.

Discussion.

The point about the kratom in the discussion made was beyond the scope and not supported by data from this study.

Feedback: Thank you for your comment and we do agree with yours. Thus, these sentences was removed from the text; “Furthermore, there has been no report on the fatalities in Southeast Asia caused by the ingestion of pure kratom alone” and “In addition, kratom use should not be reasonably expected to be safe, especially for co-use with other drugs, and it poses a public health threat.” Other than the, we also rephrase and edited some sentence in the discussion related to kratom use. 

Thank you again for your kind or comment and suggestion. We do appreciate it.

---

## [Decision Letter · Decision Letter 1]

5 Dec 2022

Patterns of Polysubstance use among Adults in Malaysia – A Latent Class Analysis

PONE-D-22-03892R1

Dear Dr. Rodzlan Hasani,

We’re pleased to inform you that your manuscript has been judged scientifically suitable for publication and will be formally accepted for publication once it meets all outstanding technical requirements.

Kind regards,

George Vousden

Staff Editor

PLOS ONE

Additional Editor Comments (optional):

Reviewers' comments:

Reviewer's Responses to Questions

**Comments to the Author**

1. If the authors have adequately addressed your comments raised in a previous round of review and you feel that this manuscript is now acceptable for publication, you may indicate that here to bypass the “Comments to the Author” section, enter your conflict of interest statement in the “Confidential to Editor” section, and submit your "Accept" recommendation.

Reviewer #1: All comments have been addressed

Reviewer #2: All comments have been addressed

2. Is the manuscript technically sound, and do the data support the conclusions?

Reviewer #1: Yes

Reviewer #2: Yes

3. Has the statistical analysis been performed appropriately and rigorously? 

Reviewer #1: Yes

Reviewer #2: Yes

4. Have the authors made all data underlying the findings in their manuscript fully available?

Reviewer #1: Yes

Reviewer #2: Yes

5. Is the manuscript presented in an intelligible fashion and written in standard English?

Reviewer #1: Yes

Reviewer #2: Yes

6. Review Comments to the Author

Reviewer #1: The authors have addressed all the comments. All required questions have been answered and that all responses meet formatting specifications.

Reviewer #2: interesting study and the informative about polysubstance and ketum use. Authors had addressed all comments made by reviewer.

7. PLOS authors have the option to publish the peer review history of their article (what does this mean?). If published, this will include your full peer review and any attached files.

Reviewer #1: **Yes: **DARSHAN SINGH

Reviewer #2: **Yes: **Mohd Azhar

---

## [Editor Report · Acceptance letter]

6 Jan 2023

PONE-D-22-03892R1 

Patterns of Polysubstance use among Adults in Malaysia – A Latent Class Analysis 

Dear Dr. Rodzlan Hasani:

I'm pleased to inform you that your manuscript has been deemed suitable for publication in PLOS ONE. Congratulations! Your manuscript is now with our production department. 

Kind regards, 

on behalf of

Dr. George Vousden 

Staff Editor

PLOS ONE